# Topical Application of Phlorotannins from Brown Seaweed Mitigates Radiation Dermatitis in a Mouse Model

**DOI:** 10.3390/md18080377

**Published:** 2020-07-22

**Authors:** Kyungmi Yang, Shin-Yeong Kim, Ji-Hye Park, Won-Gyun Ahn, Sang Hoon Jung, Dongruyl Oh, Hee Chul Park, Changhoon Choi

**Affiliations:** 1Department of Radiation Oncology, Samsung Medical Center, Seoul 06351, Korea; kyungmi.yang@samsung.com (K.Y.); kkdnsy@naver.com (S.-Y.K.); mementoamor@icloud.com (W.-G.A.); sang-hoon.jung@samsung.com (S.H.J.); dongryul.oh@samsung.com (D.O.); 2School of Medicine, Sungkyunkwan University, Seoul 06351, Korea; jh1024.park@samsung.com; 3Department of Dermatology, Samsung Medical Center, Seoul 06351, Korea

**Keywords:** radiation dermatitis, phlorotannins, mouse model, inflammation

## Abstract

Radiation dermatitis (RD) is one of the most common side effects of radiotherapy; its symptoms progress from erythema to dry and moist desquamation, leading to the deterioration of the patients’ quality of life. Active metabolites in brown seaweed, including phlorotannins (PTNs), show anti-inflammatory activities; however, their medical use is limited. Here, we investigated the effects of PTNs in a mouse model of RD in vivo. X-rays (36 Gy) were delivered in three fractions to the hind legs of BALB/c mice. Macroscopic RD scoring revealed that PTNs significantly mitigated RD compared with the vehicle control. Histopathological analyses of skin tissues revealed that PTNs decreased epidermal and dermal thickness compared with the vehicle control. Western blotting indicated that PTNs augmented nuclear factor erythroid 2-related factor 2 (NRF2)/heme oxygenase-1 (HO-1) pathway activation but attenuated radiation-induced NF-κB (nuclear factor kappa-light-chain-enhancer of activated B cells) and inflammasome activation, suggesting the mitigation of acute inflammation in irradiated mouse skin. PTNs also facilitated fast recovery, as indicated by increased aquaporin 3 expression and decreased γH2AX (histone family member X) expression. Our results indicate that topical PTN application may alleviate RD symptoms by suppressing oxidative stress and inflammatory signaling and by promoting the healing process. Therefore, PTNs may show great potential as cosmeceuticals for patients with cancer suffering from radiation-induced inflammatory side effects such as RD.

## 1. Introduction

Radiation dermatitis (RD) is one of the most common side effects of radiotherapy (RT). During or following RT, up to 95% of patients with various cancers suffer from RD [1,2]. RD exhibits a wide spectrum of symptoms and severities in the acute phase, and few of the common symptoms include erythema, dry and moist desquamation, edema, pain, pigmentation, and/or ulceration. Furthermore, RD deteriorates the patients’ quality of life [3]. Many researchers and clinicians have tried to reduce RD, but RD is still unsolved. For example, intensity-modulated RT (IMRT) can be considered as an option for less RD in breast cancer patients. Even with IMRT, however, about 30% of patients experienced desquamations [4]. Except for general skin care, the management of RD has not been established [5]. Several skin care products containing various active substances such as steroids or recombinant human epidermal growth factor (rhEGF) are commercially available and empirically prescribed in clinics [6,7,8,9].

Increasing evidence suggests that various metabolites from brown seaweeds show anti-inflammatory activities [10,11,12,13]. Among these, phlorotannins (PTNs) from brown algae, such as *Ecklonia cava*, have attracted much attention for their clinical applications, including their use as cosmeceuticals [14]. PTNs are oligomeric or polymeric phloroglucinol phenols [15], and as primary algal cell wall components, they are involved in protecting cells from ultraviolet radiation in natural environments [16,17]. Therapeutic potential of PTNs includes antioxidant, anti-inflammatory, antitumor, antiallergic, hyaluronidase inhibitory, and matrix metalloproteinases inhibitory activities [18]. The key mechanism of photo-protection by PTNs is antioxidant or redox activity [19], which supports the intrinsic cellular defense system to balance oxidative stress and reduces DNA damages. In addition, PTNs show protective effects against ionizing radiation, such as γ-rays [20]. PTNs have been reported to protect intestinal stem cells from radiation-induced damage and to facilitate the recovery of hematopoietic cells via the suppression of reactive oxygen species (ROS) production and apoptotic signaling in mice subjected to RT [21,22,23,24]. Although the anti-inflammatory effects of PTNs have been reported, their efficacy in RD management remains to be tested. In the present study, we aimed to investigate the protective effect of PTNs in a mouse model with RD.

## 2. Results

### 2.1. Development of a Mouse Model of Radiation Dermatitis

First, we developed a mouse model of RD by subjecting mouse skin to repeated irradiation of high-dose X-rays. Acute RD was scored according to the Common Terminology Criteria for Adverse Events grading criteria (Figure 1a,b) [25]. To determine the optimal radiation dose for developing the mouse model of RD, single doses of 9, 12, or 15 Gy were delivered for three consecutive days (total doses of 27, 36, or 45 Gy, respectively) to the skin of the right hind legs of anesthetized mice, and the skin reactions were monitored. RD score assessment revealed that the skin reactions first appeared 8 days after irradiation, peaked at day 17, and vanished thereafter (Figure 2a,b). The size of the radiation fractions affected the acute skin reactions. The dose of 9 Gy per fraction was much less effective and produced milder symptoms than the doses of 12 and 15 Gy per fraction (Figure 2b). There was only a slight difference in the skin reactions produced by the doses of 12 and 15 Gy per fraction. Histopathological analysis showed that epidermal and dermal thickness increased in a dose-dependent manner (Figure 2c–e). Furthermore, Masson’s trichrome staining revealed that collagen deposition increased in a dose-dependent manner (Figure 2c).

### 2.2. Efficacy of Topical Phlorotannin Application against Radiation Dermatitis

Using the mouse model of RD established by delivering three fractions of X-rays at a dose of 12 Gy per fraction, we evaluated the efficacy of topical PTN application against RD. The experimental scheme is depicted in Figure 3a. PTNs were dissolved at two different concentrations (0.05% and 0.5%) in sesame oil and topically applied on the day of irradiation. An rhEGF solution (0.005%) was used as a positive control. Upon visual assessment, both PTN treatments showed a trend of fast recovery from the point of maximum RD (Figure 3b,c). The maximum RD scores (peak values) with the 0.05% and 0.5% PTN treatments were lower than those with the vehicle control treatment (0% PTN versus 0.05% PTN, *p* < 0.001). Similarly, the RD scores on day 21 were significantly lower with the 0.05% and 0.5% PTN treatments than with the vehicle control treatment (all *p* < 0.001; Figure 3c). These results suggest that PTNs alleviate RD. The rhEGF also showed a similar trend to the PTNs, except that the maximum RD score with rhEGF treatment was similar to that with the vehicle control treatment. The lowest RD score was observed following the 0.05% PTN treatment.

### 2.3. Reduction in Radiation-Induced Epidermal and Dermal Thickening Following Topical Phlorotannin Application

Visual assessment indicated that PTN application mitigated RD in mice; therefore, histopathological analysis of the irradiated skin tissues was performed. Hematoxylin and eosin (H&E) staining showed that both the epidermis and dermis were thicker in the irradiated skin tissues than in the unirradiated ones on days 14 and 21 (Figure 4a,b). Quantification data showed that irradiation with X-rays at a dose of 12 Gy per day for three consecutive days significantly increased the epidermal (from 14.49 ± 5.33 to 181.8 ± 62.46 µm; *p* < 0.001) and dermal (from 128.8 ± 28.71 to 307.8 ± 72.72 µm; *p* < 0.001) thicknesses, whereas topical PTN application reduced this irradiation-induced epidermal and dermal thickening on day 14 (Figure 4c). The epidermal thickness was reduced to 118.5 and 95.40 µm and the dermal thickness was reduced to 203.1 and 214.2 µm with the 0.05% and 0.5% PTN treatments, respectively. Treatment with rhEGF significantly reduced the epidermal and dermal (127.7 ± 59.95 and 171.8 ± 51.09 µm, respectively; *p* < 0.001) thickness compared with the vehicle control treatment. However, on day 21, the thickness of both the layers was slightly reduced with the sham treatment (epidermis, 160.2 ± 70.42; dermis, 237.5 ± 61.52 µm), which was further reduced by PTN application (epidermis, 81.25 and 96.11 µm and dermis, 181.6 and 189.9 µm with 0.05% and 0.5% PTNs, respectively; both *p* < 0.001) but not by rhEGF application (Figure 4d). Infiltration of eosinophils dramatically increased in irradiated tissues, which was suppressed by topical treatment with both 0.05% PTNs and EGF (Appendix A). Though visual assessments indicated that 0.05% PTNs might be better than 0.5% PTNs in terms of reducing RD scores, H&E data suggest there was no difference in the thickness of the two layers following treatment with the two concentrations of PNTs and that 0.5% PTNs might be more immunosuppressive than 0.05%.

### 2.4. Modulation of NRF2, NF-κB, and AQP3 Expression Following Topical Phlorotannin Application

To elucidate the mechanism underlying the mitigating effects of PTNs on RD, we investigated nuclear factor erythroid 2-related factor 2 (NRF2) and nuclear factor-κB (NF-κB) signaling—the well-known signaling pathways related to oxidative stress and inflammation in the skin. Western blotting revealed that topical PTN application affected the expressions of NRF2, NF-κB, and their downstream targets (Figure 5a,b). On day 14 after irradiation, the expression of NRF2 and its downstream target heme oxygenase-1 (HO-1) was higher in the skin tissues topically treated with 0.05% and 0.5% PTNs than in the skin tissues in the sham and vehicle control groups (Figure 5a). On day 21, the expression levels of NRF2 and HO-1 in the PTN-treated skin tissues remained higher than those in the sham-treated tissues (Figure 5a).

The expression of NF-κB p65 in the skin tissues was increased following irradiation, which was suppressed by 0.05% and 0.5% PTN on day 14 (Figure 5b). However, PTN application induced NF-κB p65 expression in the irradiated skin tissues on day 21. The expression of cyclooxygenase 2 (COX2), which is downstream from the NF-κB signaling cascade, was also induced by radiation but suppressed by PTNs on day 14 (Figure 5b). The radiation-induced expression of COX2 was decreased on day 21. The radiation-induced expression of interleukin-1β (IL-1β) and ASC (an apoptosis-associated speck-like protein containing a caspase-recruitment domain), which are related to inflammasome activation, was suppressed by PTN treatment on day 14 (Figure 5b). On day 21, however, PTN application increased the expression of IL-1β and ASC in a dose-dependent manner. Aquaporin 3 (AQP3) expression was higher in the irradiated skin tissues than in the sham-treated tissues and further increased by 0.5% PTN on days 14 and 21 (Figure 5c). Radiation increased the phosphorylation of H2A histone family member X (γH2AX), a surrogate marker for DNA damage, which was decreased by PTN application on day 21, suggesting the rapid recovery of radiation-induced DNA damage with PTN treatment.

## 3. Discussion

In RD, the direct tissue damage caused by radiation, progression of the inflammatory response, and recovery process occur simultaneously [26]. When the skin tissue is irradiated, early damage response is initiated by highly radiosensitive cells, and the damaged epidermis may not be regenerated after repeated irradiation or prolonged radiation exposure [27]. The damaged cells release various cytokines and chemokines, which induce the inflammatory response, stimulate the growth of the surrounding blood vessels, and recruit immune cells [28]. Radiation-induced skin damage is mostly related to oxidative stress [26]. Ionizing radiation can generate ROS, which damage DNA or cellular structures, ultimately leading to the death of unrepaired cells. Antioxidant enzymes, such as superoxide dismutase, glutathione peroxidases, and thioredoxins, protect skin cells from radiation-induced oxidative stress [29]. Nevertheless, the detailed mechanism underlying RD is only partly understood. Moreover, a standard strategy for preventing or treating RD, except general skin care, remains to be established. Topical agents containing steroids or *Aloe vera* are often prescribed without evidence [30,31]. Therefore, an effective agent for the prevention and treatment of RD must be urgently developed.

Several natural substances isolated from marine algae, including PTNs, have garnered increasing attention for their medical applications [19,32,33,34,35], specifically in protecting the skin from ultraviolet radiation [33]. Although the radioprotective effects of PTNs have been tested in radiosensitive organs, including the intestine and bone marrow, these effects remain to be verified using a skin model [20]. In this context, we conducted the present study to evaluate the therapeutic efficacy of PTNs in the management of skin damage caused by RT in a mouse model. In the visual assessment, the PTN-treated groups showed lower RD scores than the control groups, even at a very low concentration, as evidenced by the lowest maximum RD scores obtained with 0.05% of concentration and significantly different time-course change from the control (Figure 3c, *p* < 0.001). Similarly, the radiation-induced increase in the thickness of the epidermis and dermis—two main layers of the skin structure—was significantly suppressed by PTN application compared to that with the control and rhEGF treatments. Together, the observed low RD scores and fast recovery indicate that PTNs likely exert a radioprotective or mitigating effect against RD.

Recent studies have demonstrated that the NRF2 pathway is implicated in both inflammatory and oxidative stress responses [36], and its modulation may be effective for managing inflammatory diseases, including dermatitis [37,38,39]. Several chemicals that can stimulate NRF2 expression have been tested for their application as therapeutic agents for RD [40]. Among natural substances, PTNs stimulate the expression of NRF2 and its downstream genes [20,41]. Our results of the western blotting of the irradiated skin tissues showed that PTNs enhanced the radiation-induced activation of the NRF2 pathway in the acute phase of RD. The visual assessment revealed that the RD scores reached near maximum around day 15 and then decreased owing to the healing process. The expression of NRF2 and HO-1 was higher in the PTN-treated groups on day 14 but not on day 21, suggesting that NRF2 is involved in the PTN-mediated mitigation of RD in the early acute phase.

In contrast, radiation induced high NF-κB p65 and COX2 expressions, which was suppressed by PTNs on day 14. The radiation-induced expression of inflammasomal proteins such as IL-1β and ASC was also suppressed by PTNs. On day 21, the suppression of inflammatory signaling was reversed in the PTN-treated groups. These data suggest that PTNs effectively suppress or delay the acute inflammatory response in the early phase of RD. Crosstalk occurs between the NRF2 and NF-κB pathways: NRF2 signaling inhibits the NF-κB pathway and vice versa [36,37]. Thus, PTNs may suppress the radiation-induced NF-κB pathway via NRF2 activation, thereby reducing the RD scores.

AQP3 is a highly abundant aquaglyceroporin in the epidermis, which is involved in hydration and can thus participate in healing and epidermal homeostasis [42]. Mice lacking this protein show dry skin and delayed wound healing [42]. A previous study has shown that AQP3 is one of the targets of NRF2 in keratinocytes during the oxidative stress response [38]. Our data showed that increased AQP3 levels were accompanied by increased NRF2 levels in PTN-treated tissues, suggesting that AQP3 is involved in the NRF2-mediated skin healing following RT.

A topical solution containing rhEGF was used as the positive control in this study. EGF is one of the best-characterized signaling growth factors, and increasing evidence suggests that EGF signaling is implicated in skin repair and inflammation [42]. Based on its potential roles, clinical trials to assess the therapeutic efficacy of EGF against diverse inflammatory diseases, including dermatitis, have been conducted. Although limited evidence is available as of now, several clinical studies have shown positive results regarding its mitigating effects on RD [6,8]. The present study demonstrated that the radioprotective effects of PTNs were comparable to those of rhEGF. However, rhEGF and PTNs likely utilize different mechanisms. EGF exerts its protective effects on skin tissues by facilitating epidermal proliferation and perturbating proinflammatory signaling, whereas PTNs likely protect cells from oxidative stress and inflammation by activating NRF2. Since EGF and PTNs were both effective against RD, combined treatment with these two metabolites is worth exploring.

This study has some limitations. First, a high radiation dose (12 Gy per fraction for three consecutive days = 36 Gy) was delivered to mice hind legs to obtain a severe RD phenotype, which may not directly reflect the clinical scenario. The typical daily radiation dose is 2 Gy per fraction. Thus, mechanisms of dermatitis and particularly cell repair change with different fractionations. Second, the visual assessment of RD may not be sufficient to compare the efficacy of PTNs and rhEGF in alleviating RD in mice. Nevertheless, our findings suggest a novel function of PTNs in mitigating RD. PTNs show great potential for use as cosmeceuticals because of their bioactive properties, including anti-inflammatory, antioxidant, anti-wrinkling, and hair growth-promoting effects [14]. Based on our results, further experimental and clinical studies are warranted to optimize the cosmeceutical formulations of PTNs and to assess their efficacy and safety in patients undergoing RT.

## 4. Materials and Methods

### 4.1. Chemicals and Reagents

PTNs were a generous gift from Won-Kyo Jung, Pukyong National University, Busan, South Korea. PTNs were prepared from *Ecklonia cava* collected along the Jeju Island coast of Korea as previously described [43]. Composition and chemical structures of PTNs in the ethanolic extracts were previously characterized [44,45]. Easyef^TM^, a commercial spray-type ointment containing rhEGF, was purchased from Daewong Pharmaceuticals (Seoul, South Korea). The primary antibodies used were as follows: anti-glyceraldehyde-3-phosphate dehydrogenase (GAPDH) (#5174), anti-p65-NF-κB (#8242), anti-IL-1β (#12242), and anti-HO-1(heme oxygenase) (#5853) purchased from Cell Signaling Technology (Danvers, MA, USA); anti-ASC (SC514414) and anti-NRF2 (SC1722) purchased from Santa Cruz Biotechnology (Dallas, TX, USA); anti-AQP3 (AB3276) purchased from Merck Millipore (Burlington, MA, USA); and anti-COX2 (#610203) purchased from BD Biosciences (Franklin Lakes, NJ, USA). Horseradish peroxidase (HRP)-conjugated secondary anti-rabbit and anti-mouse antibodies were purchased from Cell Signaling Technology.

### 4.2. Animal Experiments and Irradiation

Five-week-old female BALB/c mice were purchased from Orient Bio Animal Center (Seongnam, South Korea) and maintained under specific pathogen-free conditions under a 12-h light/12-h dark cycle. All procedures in the animal experiments were conducted in accordance with appropriate regulatory standards under the protocol reviewed and were approved by the Institutional Animal Care and Use Committee of Samsung Biomedical Research Institute at Samsung Medical Center in Seoul, South Korea (ID: 20180417001; approval date: May 8, 2018). X-ray irradiation was performed using a Varian Clinac 6EX linear accelerator (Varian, Medical Systems, Palo Alto, CA, USA) at Samsung Medical Center. Before irradiation, the mice were anesthetized via an intraperitoneal injection of 30 mg·kg^−1^ zolazepam/tiletamine and 10 mg·kg^−1^ xylazine. For the irradiation setup, all mice were placed in prone position and their whole right hind legs were fixed with tape in the irradiation field (32 cm × 7 cm) to ensure the position of irradiation area (Appendix A). The legs were placed under a 2-cm-thick water-equivalent bolus with a source-to-surface distance of 100 cm and were irradiated with 6-MV X-rays at single doses of 9, 12, and 15 Gy per fraction for three consecutive days (corresponding to total radiation doses of 27, 36, and 45 Gy, respectively) at a dose rate of 3.96 Gy per minute. The absolute X-ray dose was calibrated according to the TG-51 protocol and was verified using a Gafchromic film, with an accuracy of 1%.

### 4.3. Topical Phlorotannin Application

PTNs were freshly prepared by dissolving them at concentrations of 0.05% and 0.5% (*w*/*v*) in sesame oil (S3547; Sigma-Aldrich, St. Louis, MO, USA). Easyef^TM^ was used as a positive control. Mice hind legs were depilated using a topical cream. The irradiated skin area was covered with a PTN solution or Easyef^TM^, starting on the irradiation day. The experimental scheme is depicted in Figure 3a.

### 4.4. Measurement of Skin Tissue Damage

On days 14 and 21 after irradiation, the mice were euthanized using the gradual-fill method of CO_2_ euthanasia and the irradiated skin tissues were harvested promptly. Skin biopsies were fixed in 10% neutral-buffered formalin for 24 h, embedded in paraffin wax, and serially sectioned (thickness, 5 μm). Standard H&E and Masson’s trichrome staining protocols were used for the histological examination. Slide images were digitalized using Aperio ScanScope AT (Leica Biosystems, Buffalo Grove, IL, USA). The epidermal and dermal thickness was measured at 20 different sites in each section and averaged.

### 4.5. Western Blotting

Skin tissues were harvested on days 14 and 21 after irradiation and cut into small pieces. The frozen tissue samples were resuspended in radioimmunoprecipitation assay (RIPA) Lysis and Extraction Buffer (#89900; Thermo Fisher Scientific, Waltham, MA, USA) and lysed using TissueLyser II (QIAGEN, Hilden, Germany). The lysates were centrifuged at 13,500× *g* for 20 min at 4 °C, and the supernatants were collected. The protein concentration was determined using the Bio-Rad DC Protein Assay Kit II (Bio-Rad, Hercules, CA, USA). Equal amounts of proteins were separated via SDS-PAGE and transferred to a Protran^TM^ nitrocellulose membrane (GE healthcare, Piscataway, NJ, USA). After blocking with 10% skimmed milk for 70 min at room temperature, the membranes were incubated with primary antibodies at 4 °C overnight, followed by HRP-conjugated secondary antibodies for 1 h at room temperature. The bands of interest were visualized using an enhanced chemiluminescence detection kit (GE healthcare) according to the manufacturer’s instructions. Protein bands were quantified using ImageJ (National Institutes of Health, Bethesda, MA, USA).

### 4.6. Statistics

All data are presented as the mean ± standard deviation (SD) from three independent experiments. All statistical analyses were performed using GraphPad Prism 8.4.2 (San Diego, CA, USA). The normality of all datasets was evaluated using D’Agostino–Pearson omnibus normality test. The Brown–Forsythe test was used to test the assumption of equal variance in analysis of variance (ANOVA). For datasets with normal distribution, the comparison between groups was performed using ANOVA, followed by Bonferroni’s multiple comparison test. When the datasets were not normally distributed, the comparison between groups was performed with a Kruskal–Wallis test, followed by Dunn’s multiple test. *p* values < 0.05 were considered statistically significant.

## 5. Conclusions

The present study showed that topical PTN application alleviated RD symptoms by activating anti-inflammatory and antioxidative stress signaling in a mouse model of RD. Although RD is a common side effect in patients with cancer undergoing RT, no effective treatment regimens are available even today. Our results suggest that PTNs can be used as cosmeceuticals to prevent or alleviate skin damage during or following RT.

## Figures and Tables

**Figure 1 marinedrugs-18-00377-f001:**
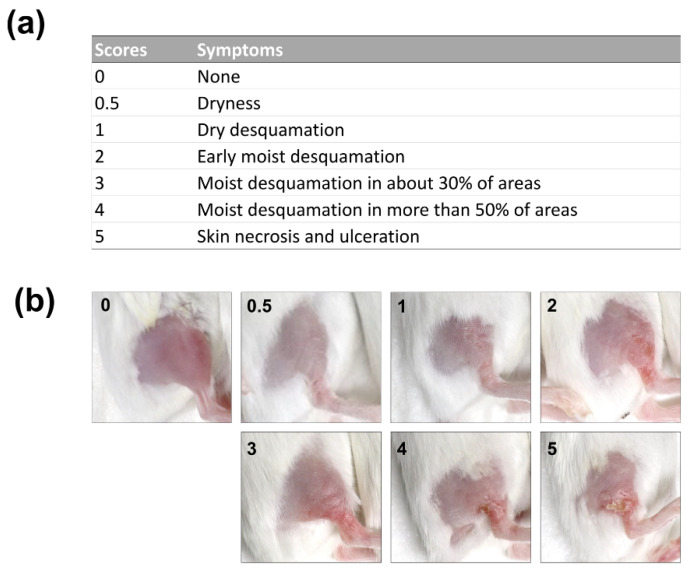
Radiation dermatitis scores in a BALB/c mouse model: (**a**) Definitions of the radiation dermatitis scores and (**b**) photographs of a mouse hind leg representing the radiation dermatitis scores.

**Figure 2 marinedrugs-18-00377-f002:**
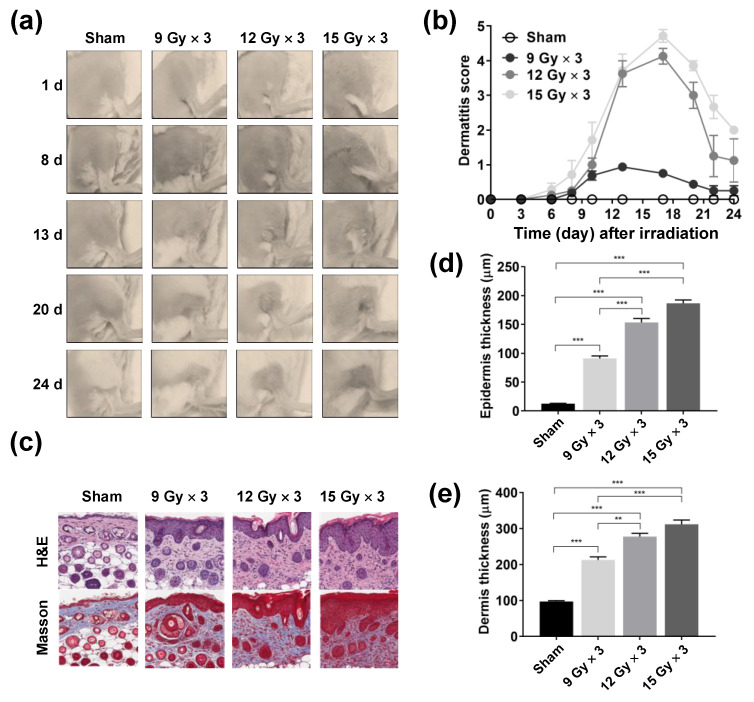
Development of the mouse model of radiation dermatitis: (**a**) Time-course changes in the skin area irradiated with various radiation doses. Photographs were obtained at the indicated time points following irradiation. (**b**) Time-dependent scoring of radiation dermatitis: Data are shown as mean ± standard deviation (SD, *n* ≥ 5 per group). (**c**) Representative immunohistochemical staining images of the irradiated skin tissue sections: hematoxylin and eosin (upper panels) and Masson’s trichrome staining (bottom panels). Skin tissues were harvested at 21 days after irradiation. (**d** and **e**) Quantification data of the epidermis (**d**) and dermis (**e**) showed dose-dependent increases in their thickness. Data are shown as mean ± SD (*n* ≥ 80 per group). Difference was evaluated using a Kruskal–Wallis test, followed by Dunn’s multiple comparison test. ** *p* < 0.01; *** *p* < 0.001.

**Figure 3 marinedrugs-18-00377-f003:**
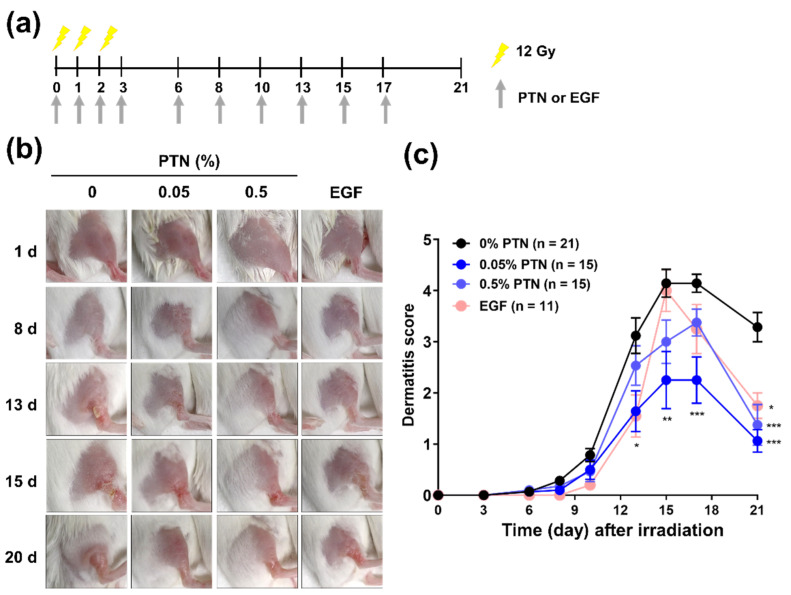
Effects of the topical application of phlorotannins (PTNs) on radiation dermatitis: (**a**) Schematic of the experimental procedures. Right hind legs were irradiated with X-rays at a dose of 12 Gy per day for three consecutive days (total, 36 Gy). PTNs were dissolved at the indicated concentrations in sesame oil and topically applied to the irradiated skin area starting on the irradiation day. An rhEGF (recombinant human epidermal growth factor) solution was used as the positive control. (**b**) Representative photographs of the irradiated skin treated with topical PTNs or rhEGF. (**c**) Time-course of changes in RD score: Data are shown as mean ± SD from three independent experiments. Difference was evaluated using one-way analysis of variance (ANOVA), followed by Bonferroni’s multiple comparison test. * *p* < 0.05; ** *p* < 0.01; *** *p* < 0.001.

**Figure 4 marinedrugs-18-00377-f004:**
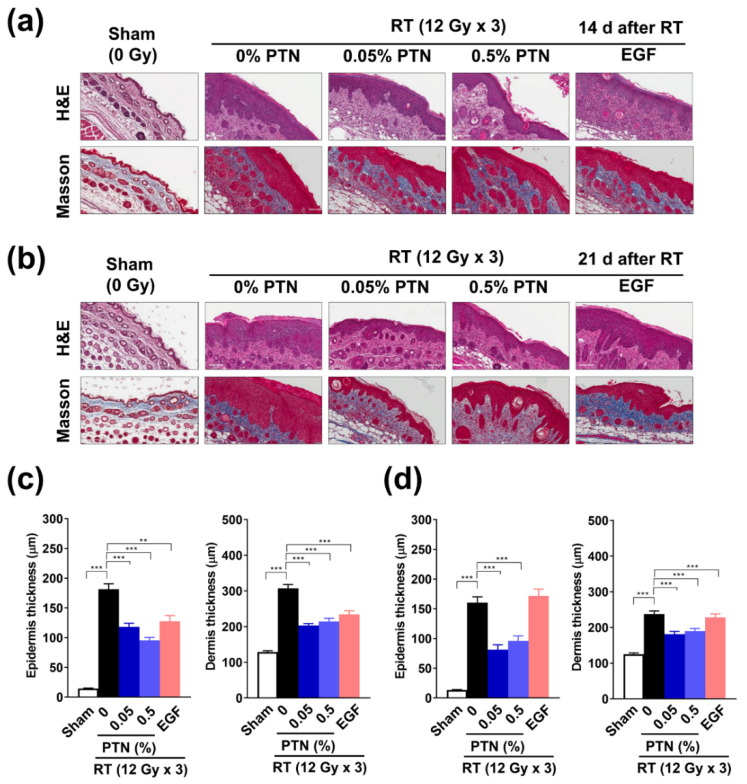
Phlorotannins (PTNs) reduce the radiation-induced thickening of the epidermis and dermis. (**a**,**b**) Representative images of hematoxylin and eosin (H&E) and Masson’s trichrome staining of the irradiated skin tissues; The skin area was topically treated with 0%, 0.05%, and 0.5% PTN or EGF. The skin tissues were collected 14 (**a**) and 21 (**b**) days after irradiation. (**c**,**d**) Measurement of the epidermal and dermal thickness in the skin tissues collected 14 (**c**) and 21 (**d**) days after irradiation: Data are shown as mean ± SD. (*n* ≥ 40). Difference was evaluated using one-way ANOVA followed by Bonferroni’s multiple comparison test. ** *p* < 0.01; *** *p* < 0.001.

**Figure 5 marinedrugs-18-00377-f005:**
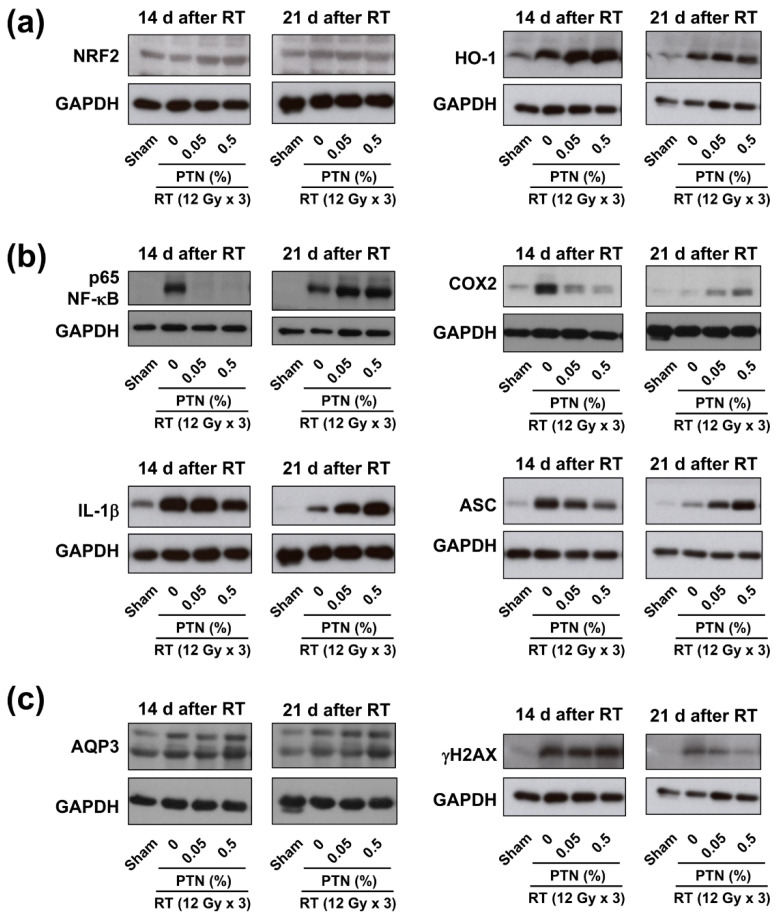
Phlorotannins (PTNs) mitigate radiation dermatitis by modulating the NRF2 and NF-κB signaling pathways: Skin tissue samples were collected 14 and 21 days after irradiation and subjected to western blotting. (**a**) Western blotting revealed that PTNs further enhanced the radiation-induced increase in NRF2/HO-1 (nuclear factor erythroid 2-related factor 2/heme oxygenase-1 pathway) expression on day 14. (**b**) PTNs attenuated the radiation-induced activation of the NF-κB pathway and inflammasome. (**c**) PTNs augmented radiation-induced aquaporin 3 (AQP3) expression but suppressed γH2AX expression in the irradiated tissues. Glyceraldehyde 3-phosphate dehydrogenase (GAPDH) was used as a loading control.

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
