# Peer review of "Topical Application of Phlorotannins from Brown Seaweed Mitigates Radiation Dermatitis in a Mouse Model"

_marinedrugs, 2020, doi:10.3390/md18080377_

Round 1

Reviewer 1 Report

Abstract line 17: RD is one of the most common side effects…

…with modern irradiation techniques (IMRT, VMAT) it is not the most common.

In the introduction (line 35) this is correct.

For my clinical point of view radiation dermatitis is an important side effect, but we have several possibilities to address it by topical medication or by avoiding it using IMRT-techniques. Therefore it is interesting to learn about anti-inflammatory effects of phlorotannins, but they will not solve a real clinical problem. Influence of phlorotannins on intestinal or lung toxicities would be much more relevant.

The fractionation of 3x12 Gy is not an adequate model for clinical use of radiation therapy (standard fractionation 5x2 Gy/week) – as the authors write themselves in the end of discussion (line 223 ff). Mechanisms of dermatitis and particularly cell repair change with different fractionations.

Phlorotannins influencing RD via ROS/oxidative stress may negatively influence the effect of irradiation against the tumor: ROS/oxidative stress are known to be most important for anti tumoral effectiveness of irradiation. In case of further development of phlorotannins in the clinical context, this will be a highly relevant question (like always in the topic of avoiding side effects by radioprotective molecules). – By similar reasons EGF has to be used with caution in tumor patients: EGFR inhibitors are used in cancer therapy and EGF would jeopardize anti tumor effects.

As my expertise is clinical radiation oncology I cannot adequately assess the lab parts of the manuscript, e.g. property of Western Plot results etc.

The results of this work contribute to a better understanding of radiation caused inflammatory effects.

Author Response

Revision details with point-by-point response to reviewers’ comments and recommendations

First, we appreciate your great reviews about our study. We read all the comments of the reviewers carefully and considered them positively to improve the quality of our research.

  1. Abstract line 17: RD is one of the most common side effects…

…with modern irradiation techniques (IMRT, VMAT) it is not the most common.

In the introduction (line 35) this is correct.

  • Thank you for the correction. We changed our abstract as you suggested. (line 17)

  1. For my clinical point of view radiation dermatitis is an important side effect, but we have several possibilities to address it by topical medication or by avoiding it using IMRT-techniques. Therefore, it is interesting to learn about anti-inflammatory effects of phlorotannins, but they will not solve a real clinical problem. Influence of phlorotannins on intestinal or lung toxicities would be much more relevant.

  • We totally agree with your opinion that IMRT can reduce radiation dermatitis in some patients. However, we think that radiation dermatitis is unresolved even with IMRT, which is used in over 50% of the patients at our center. Still, we have experienced that our patients suffered from radiation dermatitis until now, and many of them want therapeutic ointment though the effectiveness of the ointment has not been proven. It is more serious for cancers which exist close to skin or need high RT dose (i.e. head and neck cancer). In addition, patients can feel discomfort or pain even with grade 2 dermatitis in the wide skin lesion (i.e. breast cancer). Including our study, thus, many researches for radiation dermatitis are still on-going in the world. So, we added a sentence referring IMRT as a technical effort to reduce radiation dermatitis in the introduction. (line 40-42) We think that the need for our research became more evident from you.

  1. The fractionation of 3x12 Gy is not an adequate model for clinical use of radiation therapy (standard fractionation 5x2 Gy/week) – as the authors write themselves in the end of discussion (line 223 ff). Mechanisms of dermatitis and particularly cell repair change with different fractionations.

  • We agree that our fractionation regimen may be not an adequate model for clinical use in human patients, but previous studies showed that conventional fractionation dose of 2 Gy/day is not strong enough to induce dermatitis in mouse skin, partly because of high repair capacity. 12 Gy/fraction is being used in stereotactic body radiotherapy (SBRT), which means our study is still feasible. We included the limitation you pointed out in the discussion (line 262).

  1. Phlorotannins influencing RD via ROS/oxidative stress may negatively influence the effect of irradiation against the tumor: ROS/oxidative stress are known to be most important for anti tumoral effectiveness of irradiation. In case of further development of phlorotannins in the clinical context, this will be a highly relevant question (like always in the topic of avoiding side effects by radioprotective molecules). – By similar reasons EGF has to be used with caution in tumor patients: EGFR inhibitors are used in cancer therapy and EGF would jeopardize anti tumor effects.

  • The effect to tumor has to be considered, of course. However, our experiment is focused on skin toxicity. We are very sorry, but antitumor effect cannot be judged in this study. So, we decided not to cover it in this study. Also, we believe that topical agents containing very low concentration of active substances might have little effect on non-skin cancers because of skin barriers. For example, rhEGF cream is generally used and even it is considered as a cosmeceutical product. A lot of breast cancer patients use it after clinical trials referred in our study. They also have no study about this problem. What is better with phlorotainnins than EGF is that phlorotannins have an anti-tumor effect, which is explained with different pathway from EGF.

  1. As my expertise is clinical radiation oncology, I cannot adequately assess the lab parts of the manuscript, e.g. property of Western Plot results etc.

The results of this work contribute to a better understanding of radiation caused inflammatory effects.

Thank you for your in-depth review from the clinical point of view. We hope that our research will be published in this journal to help more cancer patients suffering from radiation dermatitis.

Reviewer 2 Report

The authors investigated the effects of phlorotannins in a mouse model of radiation dermatit in vivo using X-rays (36 Gy) and BALB/c mice. Hey have carried out histopathological analyses of skin tissues and found that phlorotannins decreased epidermal and dermal thickness. Using Western blot analysis, they found phlorotannins enhanced NRF2/HO-1 pathway activation but suppressed radiation-induced NF-κB and inflammasome activation, followed by mitigation of acute inflammation in irradiated mouse skin. Phlorotannins also induced fast recovery, as it is revealed by increasing of aquaporin 3 expression and decreasing of <gamma>H2AX expression. Hence, the phlorotannins from brow algae may be used for patients with radiation dermatit after cancer radiotherapy.

The article is actual and useful the conclusions correlate with experimental result, the article seems to be well written and may be published in the Marine Drugs. However, several imperfections are necessary.

  • The tested preparation should be characterized more detail and the reference on a gift is not enough. What is a certain brown algae species, i.e. the name of the plant source? What is about the certain phlorotannin fraction? How the sample was characterized, what is about chemical structures of the components of this fraction? The reference on general review [13] is not sufficient.
  • Page 2, Line 47. „phenolics” should be replaced with phenols.
  • It should be desirable to discuss the use of methyluracil against radiation dermatit for comparison because the reviewer has own personal experience in such use during radiotherapy after elimination of fibrosarcoma.

Hence, the article may be published after minor corrections.

Author Response

Revision details with point-by-point response to reviewers’ comments and recommendations

First, we appreciate your great reviews about our study. We read all the comments of the reviewers carefully and considered them positively to improve the quality of our research.

  1. The tested preparation should be characterized more detail and the reference on a gift is not enough. What is a certain brown algae species, i.e. the name of the plant source? What is about the certain phlorotannin fraction? How the sample was characterized, what is about chemical structures of the components of this fraction? The reference on general review [13] is not sufficient.

  • These are helpful comments for us. We cited the references about where and how to prepare PTNs and described them in the Materials and methods.

  1. Page 2, Line 47. „phenolics” should be replaced with phenols.

  • As you pointed, we corrected the word. (line 51)

  1. It should be desirable to discuss the use of methyluracil against radiation dermatit for comparison because the reviewer has own personal experience in such use during radiotherapy after elimination of fibrosarcoma.

  • Thank you for sharing your clinical experience. However, we decided not to add methyluracil to the discussion. The reason is that we do not have any experience with methyluracil for radiation dermatitis at all and we couldn’t find strong evidence, similarly, to empirically used steroid in clinic.

  1. Hence, the article may be published after minor corrections.

We appreciate your generous review for our study. We hope that our research will be published in this journal to help more cancer patients suffering from radiation dermatitis.

Reviewer 3 Report

  • L. 36 – “up to 95% of patients with various cancers, including skin, breast, head and neck, and lung cancers or sarcomas, suffer from RD” – there are cancers, which are not treated by radiation; therefore, these patients do not suffer from RD. Limit your claim to radiation-treated patients only.
  • L. 45 – in “phlorotannins (PTNs) in brown algae” change “in” to “from”
  • L. 49 – more detailed overview of protective effects of phlorotannins against various radiation-induced damage (including UV radiation and others) should be mentioned. These include, for example, Toxicology in vitro 23, no. 6 (2009): 1123-1130, Phytotherapy Research 22, no. 2 (2008): 238-242, Veterinary Dermatology 23, no. 1 (2012): 51-e12, or Laboratory Animal Research 21, no. 4 (2005): 385-389.
  • The last para of the Introduction should clearly postulate the hypotheses or at least the aims of the study. Avoid stating what you did. State what you aimed to do.
  • Introduction – the desired effects of phlorotannins are likely caused by matrix metalloproteinase (MMP) inhibitory activities and hyaluronidase inhibitory activity. This topic should therefore be introduced somewhere in the intro. Also the redox activity should be mentioned.
  • L. 106 – “Hematoxylin and eosin 106 (H&E) staining showed that both the epidermis and dermis were thicker in the irradiated” – The H&E stains should be supplemented with the stains for mast cells. Show, whether the content of mast cells was decreased following the treatment.
  • Fig. 4 – more concentrations of PTN need to be tested. It is obvious that both tested concentrations caused roughly the same effects. Therefore, you need to decrease the concentrations used in order to show what is the minimum concentration needed.
  • Changes in DNA damage should be quantified by, e.g., comet assay.
  • Redox status changes should be quantified by, e.g., DCFH-DA.
  • L. 184 – “seemed effective” – all the presented data should be subjected to rigorous statistical analyses. Any reported increase/decrease should be accompanied with relevant statistical test.
  • Fig. 4 – when outcomes of some statistical tests were already shown, such as in this figure, they were incorrectly reported. It is necessary to report the type of the test used, n, test outcome and not just the p value.
  • L. 234 – disclose analytical data concerning the purity of the PTN used.
  • Chapter 4.2 – describe in a clearer way how did you ensure the size and position of the irradiated area.
  • L. 288 – you indicate the use of ANOVA; however, you do not indicate, whether the data distribution normality and variance equality were tested and with which results.
  • L. 339 – use italics for Latin terms, such as “Cystoseira usneoides”
  • L. 296 – there is a chapter entitled “Supplementary materials” but no supplementary materials were provided. Having raw data available would be of interest.
  • Explain why you avoided the use of the positive control in Fig. 5
  • Double-check the use of abbreviations, such as the abbreviation of “aquaporin”. When introduced, they should be consistently used throughout the text.
  • L. 156 – “Skin aquaporin 3 (AQP3) is involved in hydration and can thus participate in healing and epidermal homeostasis [22].” – avoid references and discussion-like text in the Results chapter. Move it to the Discussion.
  • L. 207 – “suppress the radiation-induced NF-B pathway via NRF2 activation, thereby reducing the RD scores. AQP3 is a highly abundant aquaglyceroporin in the epidermis” – make sure that the text is fluent and that the sentences are linked to one another. Above, I copy-pasted the example of two sentences, which are not linked. The topic of the second sentence (AQP3) is unrelated to the topic of the previous sentence.
  • L. 173 – what are “Empirical topical agents”? Rephrase.

Author Response

Revision details with point-by-point response to reviewers’ comments and recommendations

  1. L. 36 – “up to 95% of patients with various cancers, including skin, breast, head and neck, and lung cancers or sarcomas, suffer from RD” – there are cancers, which are not treated by radiation; therefore, these patients do not suffer from RD. Limit your claim to radiation-treated patients only.

  • First, we appreciate your detailed review.
  • In the first paragraph of the comments, we guess that you meant the radiotherapy as a definitive treatment. Using radiotherapy in clinics, we treat the cancers mentioned above for various treatment options (i.e. postoperative, palliative, or neoadjuvant settings). For clear meaning, nevertheless, we added one more review paper as a reference and removed specific name of cancers as your comment. (line 36)

  1. L. 45 – in “phlorotannins (PTNs) in brown algae” change “in” to “from”

  • Thank you for the correction. We changed the word in the manuscript. (line 49)

  1. L. 49 – more detailed overview of protective effects of phlorotannins against various radiation-induced damage (including UV radiation and others) should be mentioned. These include, for example, Toxicology in vitro 23, no. 6 (2009): 1123-1130, Phytotherapy Research 22, no. 2 (2008): 238-242, Veterinary Dermatology 23, no. 1 (2012): 51-e12, or Laboratory Animal Research 21, no. 4 (2005): 385-389.

  • As you recommended, we cited the studies and described the mechanism of photo-protection. (line 53)

  1. The last para of the Introduction should clearly postulate the hypotheses or at least the aims of the study. Avoid stating what you did. State what you aimed to do.

  • We corrected the sentence which mentioned the purpose of this study. (line 61)

  1. Introduction – the desired effects of phlorotannins are likely caused by matrix metalloproteinase (MMP) inhibitory activities and hyaluronidase inhibitory activity. This topic should therefore be introduced somewhere in the intro. Also the redox activity should be mentioned.

  • The activities of phlorotannins were added in the introduction part with another reference. (line 53)

  1. L. 106 – “Hematoxylin and eosin 106 (H&E) staining showed that both the epidermis and dermis were thicker in the irradiated” – The H&E stains should be supplemented with the stains for mast cells. Show, whether the content of mast cells was decreased following the treatment.

  • Thanks for the helpful suggestion. Instead of mast cells, we counted eosinophils in the H&E stained tissue sections and found infiltration of eosinophils significantly decreased following the PTN treatment. We included these data as Figure S2.

  1. Fig. 4 – more concentrations of PTN need to be tested. It is obvious that both tested concentrations caused roughly the same effects. Therefore, you need to decrease the concentrations used in order to show what is the minimum concentration needed. Changes in DNA damage should be quantified by, e.g., comet assay. Redox status changes should be quantified by, e.g., DCFH-DA.

  • We tested two different concentrations of PTNs, and their effects did not look same, except for some data. Treatment with 0.05% PTNs was more effective than 0.5% in terms of reducing maximum RD when compared to vehicle control (Fig. 3C; p < 0.001). Despite little difference in epidermal/dermal thickness between the two groups, inflammatory signaling was more suppressed by 0.5% PTNs than 0.05% as judged by western blots and infiltration of eosinophils. We do not know what minimum and maximum concentrations are required for RD mitigation at this time, but our data suggest both concentrations of PTNs are effective and they work differently. We agree that comet and DCF-DA assays are useful to quantify DNA damage and intracellular ROS level, respectively. However, these assays are more likely to be powerful at experiments using single cells such as keratinocytes and our study focused on in vivo setting. Thus, we believe that expression of gamma-H2AX and NRF2/HO1 in tissue samples may be biologically relevant enough.

  1. L. 184 – “seemed effective” – all the presented data should be subjected to rigorous statistical analyses. Any reported increase/decrease should be accompanied with relevant statistical test.

  • Thanks for the helpful comments. We reanalyzed maximum RD scores between groups and displayed statistic significance in the Fig. 3c. We edited the sentence, accordingly. (line 205)

  1. Fig. 4 – when outcomes of some statistical tests were already shown, such as in this figure, they were incorrectly reported. It is necessary to report the type of the test used, n, test outcome and not just the p value.

  • We revised the figure legends of Fig. 2, 3, and 4 that now include more detailed information such as the types of the statistical test and data numbers.

  1. L. 234 – disclose analytical data concerning the purity of the PTN used.

  • We cited the references describing analytical data of PTNs used in this study in the Materials and methods.

  1. Chapter 4.2 – describe in a clearer way how did you ensure the size and position of the irradiated area.

  • All mice were placed in prone position and their whole right hind legs were fixed with tape in the irradiation field to ensure the position of the irradiation area. We included Figure S1 demonstrating how we set up irradiation experiments and revised Chapter 4.2 accordingly.

  1. L. 288 – you indicate the use of ANOVA; however, you do not indicate, whether the data distribution normality and variance equality were tested and with which results.

  • Thanks for the comments. We reanalyzed the data distribution normality and variance equality of all our datasets and displayed the statistic results in the figure legends. We also revised 4.6 statistics accordingly.

  1. L. 339 – use italics for Latin terms, such as “Cystoseira usneoides”

  • All Latin terms used italics in the revised manuscript including the references.

  1. L. 296 – there is a chapter entitled “Supplementary materials” but no supplementary materials were provided. Having raw data available would be of interest.

  • We included Figure S1 and S2 as supplementary materials in the revised manuscript.

  1. Explain why you avoided the use of the positive control in Fig. 5

  • The purpose of this study is not investigating the effect of EGF on radiodermatitis, which has been already reported in many literatures.

  1. Double-check the use of abbreviations, such as the abbreviation of “aquaporin”. When introduced, they should be consistently used throughout the text.

  • The abbreviations were checked.

  1. L. 156 – “Skin aquaporin 3 (AQP3) is involved in hydration and can thus participate in healing and epidermal homeostasis [22].” – avoid references and discussion-like text in the Results chapter. Move it to the Discussion.

  • The sentence introducing aquaporin 3 moved to the discussion. Thank you for the arrangement.

  1. L. 207 – “suppress the radiation-induced NF-kB pathway via NRF2 activation, thereby reducing the RD scores. AQP3 is a highly abundant aquaglyceroporin in the epidermis” – make sure that the text is fluent and that the sentences are linked to one another. Above, I copy-pasted the example of two sentences, which are not linked. The topic of the second sentence (AQP3) is unrelated to the topic of the previous sentence.

  • Thanks for the comments. We divided the paragraph in two paragraphs.

  1. L. 173 – what are “Empirical topical agents”? Rephrase.

  • “Empirical” and “without evidence are overlapped. So, we removed one and edited the sentence. (line 196)

We believe that your comments improved the quality of this study, though it still has limitations. We hope that our research will be published in this journal to help more cancer patients suffering from radiation dermatitis.

Round 2

Reviewer 3 Report

The provided comments were largely reflected. I would welcome would the authors attempt to focus much more on the mechanisms of the observed in vivo effects, but I believe that they will do it at least in their future experiments. Also the effects of lower concentrations of the active compound need to be explored.